# Comparative Neurological and Behavioral Assessment of Central and Peripheral Stimulation Technologies for Induced Pain and Cognitive Tasks

**DOI:** 10.3390/biomedicines12061269

**Published:** 2024-06-06

**Authors:** Muhammad Danish Mujib, Ahmad Zahid Rao, Muhammad Abul Hasan, Ayesha Ikhlaq, Hira Shahid, Nargis Bano, Muhammad Usman Mustafa, Faisal Mukhtar, Mehrun Nisa, Saad Ahmed Qazi

**Affiliations:** 1Department of Biomedical Engineering, NED University of Engineering & Technology, Karachi 75270, Pakistan; ahmadrao@neduet.edu.pk (A.Z.R.); abulhasan@neduet.edu.pk (M.A.H.); 2Neurocomputation Lab, National Centre of Artificial Intelligence, NED University of Engineering & Technology, Karachi 75270, Pakistan; shahidh8@uni.coventry.ac.uk (H.S.); saadqazi@neduet.edu.pk (S.A.Q.); 3Institute of Physics, The Islamia University of Bahawalpur, Bahawalpur 63100, Pakistan; ayesha.ikhlaq@iub.edu.pk (A.I.); usman.mustafa@iub.edu.pk (M.U.M.); faisal.mukhtar@iub.edu.pk (F.M.); 4Research Centre for Intelligent Healthcare, Coventry University, Coventry-CV1 2TU, UK; 5Department of Physics and Astronomy College of Science, King Saud University, P.O. Box 2455, Riyadh 11451, Saudi Arabia; 6Department of Physics, Govt. Sadiq College Women University, Bahawalpur 63100, Pakistan; mehr.phy@gscwu.edu.pk; 7Department of Electrical Engineering, NED University of Engineering & Technology, Karachi 75270, Pakistan

**Keywords:** pain, physiology, data analysis, signal transduction, transcranial direct current stimulation, transcutaneous electrical nerve stimulation, Stroop task, cognition, feature extraction

## Abstract

Pain is a multifaceted, multisystem disorder that adversely affects neuro-psychological processes. This study compares the effectiveness of central stimulation (transcranial direct current stimulation—tDCS over F3/F4) and peripheral stimulation (transcutaneous electrical nerve stimulation—TENS over the median nerve) in pain inhibition during a cognitive task in healthy volunteers and to observe potential neuro-cognitive improvements. Eighty healthy participants underwent a comprehensive experimental protocol, including cognitive assessments, the Cold Pressor Test (CPT) for pain induction, and tDCS/TENS administration. EEG recordings were conducted pre- and post-intervention across all conditions. The protocol for this study was categorized into four groups: G1 (control), G2 (TENS), G3 (anodal-tDCS), and G4 (cathodal-tDCS). Paired *t*-tests (*p* < 0.05) were conducted to compare Pre-Stage, Post-Stage, and neuromodulation conditions, with t-values providing insights into effect magnitudes. The result showed a reduction in pain intensity with TENS (*p* = 0.002, t-value = −5.34) and cathodal-tDCS (*p* = 0.023, t-value = −5.08) and increased pain tolerance with TENS (*p* = 0.009, t-value = 4.98) and cathodal-tDCS (*p* = 0.001, t-value = 5.78). Anodal-tDCS (*p* = 0.041, t-value = 4.86) improved cognitive performance. The EEG analysis revealed distinct neural oscillatory patterns across the groups. Specifically, G2 and G4 showed delta-power reductions, while G3 observed an increase. Moreover, G2 exhibited increased theta-power in the occipital region during CPT and Post-Stages. In the alpha-band, G2, G3, and G4 had reductions Post-Stage, while G1 and G3 increased. Additionally, beta-power increased in the frontal region for G2 and G3, contrasting with a reduction in G4. Furthermore, gamma-power globally increased during CPT1, with G1, G2, and G3 showing reductions Post-Stage, while G4 displayed a global decrease. The findings confirm the efficacy of TENS and tDCS as possible non-drug therapeutic alternatives for cognition with alleviation from pain.

## 1. Introduction

Pain is a multisystem disorder that influences psychological and neurological processes resulting in discomfort by evoking a maladaptive behavior reinforced by inadequate treatment and misuse of medication [1]. Pain due to activation of afferent nerve fibers in response to noxious thermal, chemical, or mechanical stimuli with potential to cause threatening damage to non-neural tissue is characterized as nociceptive pain [2,3] and can be induced by multiple approaches, some of which include the Peltier device, the Cold Pressor Test, and the pin-prick test [4,5,6,7].

The interaction between pain and cognitive processes, notably attention, has gained increasing interest. Pain serves as an alarming signal and indicates an inverse relationship with a stimulus related to a cognitive response [8]. Within the psychosocial context, the cognitive component of pain represents the integration of different components, including attention, working memory, anticipation, and short-term and long-term memory [9]. It has been reported in a meta-analysis that in healthy individuals, laboratory (experimentally)-induced pain degrades the performance of attention-demanding tasks by directing it towards pain location [10,11,12]. The attention-grabbing nature of pain highlighted in the literature overstresses cognitive networks, especially in response to a pain-related sensory stimulus, which diminishes the overall cognitive performance, most notably attention [13,14]. Considering the inverse relationship between pain and cognitive stimuli, pain perception can be inhibited by engagement of cognitive tasks such as the N-back task, Stroop task, visual span, digit span, and attention network task, thereby ensuring the optimal performance of cognitive functions such as, response time, working memory, and attention in the presence of pain distractors [13,15,16,17].

In order to enhance impaired cognitive functions and pain inhibition, a variety of non-pharmacological neuromodulation (central and peripheral stimulation) modalities are available, some of which include Repetitive Transcranial Magnetic Stimulation (rTMS), Transcranial Direct Current Stimulation (tDCS), and Transcutaneous Electrical Nerve Stimulation (TENS). TENS, a peripheral nerve stimulation technique, is widely known for its analgesic effects produced by the activation of multiple peripheral and central inhibitory mechanisms and is effective for both acute and chronic pain conditions [18,19,20,21,22]. In addition to its hypoalgesic effect, it enhances cognitive and executive functions by activating brain regions associated with cognitive processing [23,24,25].

tDCS, through anodal and cathodal stimulation, offers a promising solution for managing neurological and psychological conditions by modulating neuronal functionality in targeted brain regions [26,27,28]. Anodal-tDCS has been shown to increase excitability in the cortex, whereas cathodal-tDCS has been shown to decrease excitability, highlighting its potential therapeutic effects [29]. It is evident from prior studies that the dorsolateral prefrontal cortex (DLPFC) is responsible for the circuits of pain [30], including the attentional circuit dedicated to noxious stimuli [31]. Previous studies performed in healthy volunteers showed that stimulation of the DLPFC modulates cognitive functions by enhancing pain inhibition and reducing pain perception for experimentally induced pain [32,33].

Relief in cold pain perception is also noticed when anodal-tDCS is applied to the left primary motor cortex [34]. An escalated behavioral performance represented by higher efficiency and efficacy in cognitive processes is also observed when the left prefrontal cortex is stimulated with anodal-tDCS [35,36].

Studies conducted previously on pain inhibition and cognitive enhancement show promising effects on neuromodulation modalities after the application of TENS and tDCS [32,37,38], but comparisons between them are lacking. No evidence regarding the effectiveness of central stimulation over peripheral stimulation has been reported in previous studies. Former studies investigating pain inhibition via peripheral nervous system activation with TENS lack cognitive analysis, while those examining cognitive changes have not considered experimentally induced pain in healthy subjects.

In light of this gap in the literature, the primary aims of this study were to compare the effectiveness of central stimulation (tDCS) and peripheral stimulation (TENS) in pain inhibition during a cognitive task in healthy volunteers and to observe potential neurological and cognitive improvements. This study specifically applied tDCS to the DLPFC over F3/F4 (anodal-tDCS: anode-F3, cathode-F4; cathodal-tDCS: cathode-F3, anode-F4) [39,40] and TENS to the median nerve [41,42]. The Cold Pressor Test (CPT) was used to induce thermal pain [43,44]. Participants self-reported pain intensity and pain tolerance. Cognitive performance was also measured with the Stroop test [45,46,47].

## 2. Materials and Methods

### 2.1. Participants

A total of 80 healthy participants (46 males and 34 females; 20–30 years (25.20 ± 2.06 years)) were recruited. Participants were screened to ensure they were not experiencing pain at the time of recruitment, and those with other general diseases were excluded. Also, the participants under the effect of analgesics, antihypertensives, and others were also excluded. The protocols of the study were explicitly explained to all the participants, and an informed consent form was voluntarily signed and submitted by the participants before commencement of the experiments. The study was conducted with the approval of the Research Ethic Committee, NED University of Engineering and Technology.

### 2.2. Experimental Procedure

The procedure was explained to the subjects, instructing them to sit down and relax. EEG recordings were conducted while participants were in a state of relaxed eye-closure for a duration of 5 min. Subsequently, the Stroop task, a cognitive assessment, was administered. Following completion of the cognitive task, participants underwent the Cold Pressor Test (CPT) to measure pain intensity and tolerance. Upon completion of the CPT, participants received either anodal-tDCS, cathodal-tDCS, or TENS stimulation, lasting for 15 min. This comprised 10 min of sham stimulation, followed by an additional 5 min of active stimulation, allowing participants to acclimate to the sensation. Following the conclusion of the stimulation sessions, EEG recordings were once again obtained during a 5-min period with eyes closed. In conclusion, participants performed the cognitive test followed by the CPT. Brief breaks were permitted between sub-sessions.

The experimental procedure in this study involved categorizing participants into four groups, ensuring an equal distribution of participants across each group. In the behavioral and psychological assessment, there were 20 participants in each group. However, in the neurological assessment, the number of participants in each group was limited to 12. Among the four groups, one was the control group named G1, and this group did not receive any neuromodulation, whereas electrical stimulation via TENS, anodal-tDCS, or cathodal-tDCS was administered to each of the remaining groups, namely, G2, G3, and G4, respectively. The stimulation via tDCS was administered to the dorsolateral prefrontal cortex (F3/F4), whereas electrical stimulation via TENS was delivered to the medial nerve. Each group underwent the seven stages as shown in Figure 1.

### 2.3. Cold Pressor Test for Pain Induction

The Cold Pressor Test is a preferred technique for inducing nociceptive thermal pain. The experiment commenced in accordance with the guidelines recommended in prior studies [4,48,49]. The Cold Pressor Test was conducted in a quiet room with no visual or auditory distractions. The participants were comfortably seated in a convenient room with a temperature of 26 ± 2 °C. Dometic Cool Fun CK 40D Hybrid (Mobicool, Zhuhai, Guangdong, China) was employed, with the power of compressor cooling technology, and conveniently adjusted the temperature of water at 1 °C to 3 °C. The participants were instructed to submerge their dominant hand up to their wrist in icy cold water, and a stopwatch was simultaneously activated to record the pain threshold, which was the time latency to the initiation of pain sensation reported by each participant. The time at which the participants withdrew their hand from the chilled water, when the intensity of pain became unbearable, was also recorded and marked as their pain tolerance.

### 2.4. Transcranial Direct Current Stimulation

Transcranial Direct Current Stimulation (tDCS) current was applied using a constant current stimulator (Brain Driver, Chicago, IL, USA). A pair of surface electrodes with a surface area of 20 cm^2^ was employed for the administration of direct current. These surface electrodes were soaked in saline solution to enhance conductivity. In all, 10–20 international EEG system electrodes were placed over the DLPFC, which is the preferred brain area for inducing electrical stimulations. Anodal-tDCS stimulation was delivered with an anode positioned over F3 and a cathode placed over the F4. This electrode placement was reversed for administration of cathodal-tDCS stimulation, with a cathode placed over F3 and an anode placed over F4. An offline stimulation of 2 mA intensity was administered for 5 min indicating that tDCS was not applied during the psychometric task of cognition and pain.

### 2.5. Transcutaneous Electrical Nerve Stimulation

Transcutaneous electrical nerve stimulation (TENS) was delivered using a portable battery-powered TENS stimulator (InTENSity Select Combo II, St. Louis, USA). A symmetrical biphasic rectangular wave TENS signal was administered via a pair of self-adhesive surface electrodes of 5 cm × 5 cm. The electrodes were strategically positioned over the median nerve, which governs sensations in the dominant hand subjected to thermal pain induction via CPT. They were specifically placed on the wrist, positioned above the hand immersed in cold water [42,50]. An output frequency of 100 Hz, an electrode impedance (below 2 KΩ) and pulse width of 100 microseconds, and a current intensity of 10 mA were delivered for 5 min.

### 2.6. EEG Data Acquisition

For the purpose of acquiring real time EEG data, an Emotive Epoc device (Emotive Inc., San Francisco, CA, USA) with a sampling frequency of 128 Hz was employed. Emotive Epoc is a wireless EEG headset comprising 14 channels and saline-based electrodes. The location of the 14 electrodes of the Emotive EPOC were in accordance with the 10–20 international system of EEG montage, with the electrodes placed at AF3, AF4, F3, F4, F7, F8, FC5, FC6, P7, P8, T7, T8, O1, and O2.

### 2.7. EEG Data Analysis

EEG signal processing was conducted offline, for which a fifth-order high-pass IIR filter was set to 1 Hz, and a notch filter of 49.5 to 50.5 Hz by using (1) [51] was employed to eliminate noise from the line frequency.
(1)$$y\left(m\right)=\sum_{k=1}^{N}a_{k}y\left(m-k\right)$$

The Welch method was applied for the estimation of spectral density by using (2) [52].
(2)$$P_{welch}\left(f\right)=\frac{1}{k}\sum_{k=0}^{k-1}{\left|FFT\left[x_{k}\left[n\right]\right]\right|}^{2}$$

Here, *k* is the number of segments and $X_{k}(f)$ is the Fourier transform of the *k*-th segment. The *FFT* is typically used to efficiently compute the Fourier transform.

The sampled data are segmented to improve spectrum estimation. In order to conserve the quality of the EEG and reduce the power spectral variance, a delicate balance is required between the segment length and the overlap rate. In the current study, a Hanning window of length 512 samples corresponded to the data of 4 s with a 50% overlap to avoid spectral variance. A resolution of 0.25 was achieved with the Welch method. Therefore, the mean of the power spectrum (3) [53] was computed across five frequency bands: delta (1–4 Hz), theta (4–8 Hz), alpha (8–12 Hz), beta (13–30 Hz), and gamma (30–40 Hz).

If the PSD estimate $P_{welch}(f)$ for each frequency bin or band was achieved, the mean power spectrum $\left(\bar{P}(f)\right)$ was computed as follows:


(3)
$$\bar{P}(f)=\frac{1}{N}\sum_{i=1}^{N}P_{welch,i}(f)$$


### 2.8. Behavioral Assessment

#### 2.8.1. Numeric Pain Scale

The pain intensity was rated using the Numeric Pain Scale ranging from 0 to 10, where 0 represents no pain and 10 corresponds to extremely severe pain [54] as shown in Figure 2. During CPT, participants mark the number that represents their pain level.

#### 2.8.2. Stroop Task

For the purpose of evaluating attentiveness, the Stroop task was performed for 6 min. A string of a consistent color word and an inconsistent color word and geometric shapes were presented on a screen as the stimulus. The participants were instructed to press the right arrow key on the keyboard if the color of the geometric shape color and the color of the word matched either diagonally or in parallel; otherwise, they were instructed to press the left arrow key on the keyboard, as shown in Figure 3.

The stimuli stayed on the computer screen for 3 s, and the task participant was then allowed to respond. Since two cognitive tasks are being processed by the brain simultaneously, this leads to an increase in cognitive load, and the response times of the participants vary. This helps to determine the cognitive speed, attentional capacity, and response time of the individual. The test was administered pre- and post-application of tDCS. The Stroop test score was calculated by using (4).


(4)
$$\mathrm{Stroop}\_\mathrm{Score}=\frac{Correct\_Response}{Total\_Response}\times 100$$


Here, *Correct_Response* is the number of correct responses given by the participant during the Stroop test.

*Total_Responses* is the total number of trials attempted by the participant.

#### 2.8.3. Statistical Analysis

A significant effect of neuromodulation on pain tolerance, pain inhibition, and cognitive test results (the Stroop test score) was evaluated using the pairwise *t*-test for different stimulation types (anodal-tDCS, cathodal-tDCS, TENS, and control) and stimulation condition (Pre-Stage and Post-Stage). The t-values were calculated by using (5) for Pre-Stage, Post-Stage, and neuromodulation conditions All results are expressed and stated as the mean ± standard error of the mean, and *p* < 0.05 was set as the significance level.
(5)t=A¯1−A¯2SDNp

Here, ${\bar{A}}_{1}$ and ${\bar{A}}_{2}$ are the sample means of the two conditions (Pre-Stage and Post-Stage).

*SD* is the standard deviation of the difference between the paired observations.

*Np* is the number of paired observations.

The Cohen method was utilized to determine the effect size and ascertain whether the training-induced changes hold practical significance, thereby ensuring against false positives in behavioral alterations within the Post-stage [7]. The effect size was computed for the Stroop test score, pain intensity, and pain tolerance. This involved subtracting the mean Stroop test scores of the two groups and dividing by the pooled standard deviation (refer to (6)). Likewise, the effect size was determined for the mean and standard deviation of the pain intensity and pain tolerance across group pairs employing the same formula (Equation (6)). The effect sizes exceeding 0.8 were deemed large, while those falling between 0.4 and 0.8, as well as those below 0.4, were categorized as medium and low effect sizes, respectively [7].
(6)$$Cohen\;d=\frac{{\bar{M}}_{1}-{\bar{M}}_{2}}{\sqrt{\frac{{SD}_{1}\left(n_{1}-1\right)+{SD}_{2}(n_{2}-1)}{n_{1}+n_{2}-2}}}$$
where ${\bar{M}}_{1}$ and $M_{2}$ are the mean values, ${SD}_{1,2}$ are the standard deviations, and $n_{1,2}$ are the sample sizes of two variables.

## 3. Results

Initially, we performed an interim analysis on the first seven participants per group (total 28 participants) to calculate the effect size and statistical power for the groups showing significant changes in cognitive score, pain intensity, and pain tolerance level. G2 displayed significant changes in both pain intensity (d = 1.21, SP = 76.01%) and pain threshold (d = 1.72, SP = 96.33%), whereas G4 underwent significant changes in pain intensity (d = 1.48, SP = 90.04%) and pain tolerance (d = 1.56). G3 displayed a significant change in Stroop test scores (d = 1.49, SP = 90.4%). Based on the statistical power analysis and these above effect sizes for each group, a maximum sample size of 19 participants per group (total 76) was required to conduct this study when *p* < 0.025 for SP = 90%. We used the first 28 participants for both the interim and final analyses. Therefore, to avoid false positives due to this multiple comparison, we set *p* < 0.025 rather that *p* < 0.05.

Analysis of demographic data revealed no statistically significant difference (*p* > 0.05) in age across all four groups (G1 = 25.1 ± 2.12 years, G2 = 25.6 ± 2.76 years, G3 = 25.5 ± 2.17 years, G4 = 24.9 ± 2.33 years). Each group comprised 20 participants, with 6 females in groups G1 and G4 and 5 females in G2 and G3.

All 80 participants were involved in the present study. However, 32 participants chose not to wear the EEG device (Emotive Inc., San Francisco, US) during the entire experiment due to discomfort but still completed the experiment without wearing the device. The 48 remaining participants, with 12 in each group, completed the experiment wearing the EEG device. They received both tDCS and TENS electrical stimulations and underwent experimentally induced pain via CPT, with no persistent negative effects noted.

### 3.1. Psychological Analysis

#### 3.1.1. Effects of Electrical Stimulation on Pain Perception

Figure 4 demonstrates the effects of neuromodulation on pain perception (i.e., pain intensity and pain tolerance) for the CPT1 (black bars) and CPT2 pain assessments (gray bars) for all four groups (G1, G2, G3, and G4). Sub-figure (a) shows the mean pain intensity level for the participants in both CPT1 and CPT2, while sub-figure (b) shows the mean pain tolerance level for the participants in both CPT1 and CPT2. Also, the significant changes in the pain intensity and pain tolerance level in CPT2 as compared to CPT1 are marked by an asterisk (*) at the top of each bar.

The outcomes for pain perception for neuromodulation modalities of tDCS (anodal and cathodal) and TENS were compared to determine their effectiveness using the visual analog scale. The pain intensity scores of TENS (G2) (*p* = 0.002, t-value = −5.34), anodal-tDCS (G3) (*p* = 0.061), and cathodal-tDCS (G4) (*p* = 0.023, t-value = −5.08) exhibited a significant difference when compared to the control group (G1). The maximum efficacy was obtained by administering TENS to the median nerve with a pain tolerance of *p* = 0.009, t-value = 4.98. Cathodal-tDCS (G4) to the DLPFC also indicated an augmented endurance to pain with a tolerance value of *p* = 0.001, t-value = 5.78. However, application of anodal-tDCS to the DLPFC revealed a moderate effect on pain tolerance with *p* = 0.082.

#### 3.1.2. Influence of Electrical Stimulation on the Stroop Test Task

Figure 5 demonstrates the effects of neuromodulation on a cognitive task (i.e., Stroop test) for Stroop Test1 (black bars) and Stroop Test2 cognitive assessments (gray bars) for all four groups (G1, G2, G3, and G4). The figure shows the mean Stroop test score obtained for the participants in both Stroop Test1 and Stroop Test2. Also, the significant changes in the Stroop test score for Stroop Test2 as compared to Stroop Test1 are marked by an asterisk (*) at the top of each bar. To examine the effectiveness of non-pharmacological central and peripheral stimulants on cognition, particularly attention, the Stroop test was performed. An intra-group comparison revealed a significant difference in cognitive performance, with a reduction in omission and wrong replies, indicating an overall enhancement; this finding was most evident with administration of anodal-tDCS with a *p* value of 0.041 and t-value = 4.86. A moderate beneficial effect was observed with TENS and cathodal-tDCS with *p* = 0.059 and 0.067, respectively.

### 3.2. Neurological Analysis

In this study, EEG recordings were conducted in a resting state under both eyes-open and eyes-closed conditions across four different groups, with Group 1 serving as the control. Utilizing *t*-tests for comparison, our analysis revealed no statistically significant differences in Pre-Stage EEG results for all four groups. Figure 6 shows the EEG power for G1, G2, G3, and G4 within the delta-band.

The first row shows the mean power of the Pre-Stage, CPT1, and Post-Stage EEG for each group. The second row shows a significant difference between the Pre-Stage and CPT1 as well as post-stages across all groups. The third row shows the significant difference between CPT1 vs. Post-Stages for all groups. Black circles indicate reductions in significant power, and grey circles indicate increases. Specifically, there was a significant global increase in the delta power in the CPT1 condition, and a significant increase only in the frontal region was observed in the Post-Stage of G3 compared to the Pre-Stage EEG. Conversely, there was a significant decrease in the delta power in the Post-Stage EEG of G2 and G4 compared to the Pre-Stage. When compared to the CPT1 condition, G2 and G4 significantly decreased in the frontal region, while G3 increased in the parietal and occipital regions.

Figure 7 shows the EEG power of G1, G2, G3, and G4 within the theta-band. The first row shows the mean power of the Pre-Stage, CPT1, and Post-Stage for each group. The second row shows a significant difference between the Pre-Stage and CPT1 as well as Post-Stage across all groups. The third row shows a significant difference between CPT1 vs. Post-Stage for all groups. Black circles indicate reductions in significant power, and grey circles indicate increases. Specifically, there was a significant increase in the theta power in the occipital region in the CPT1 condition and Post-Stage of G2, while no significant changes were observed in other groups as compared to Pre-Stage. When compared to the CPT1 condition, G1 and G3 significantly decreased in the parietal region, while G4 increased in the parietal and occipital regions.

Figure 8 shows the EEG power for G1, G2, G3, and G4 within the alpha-band. The first row shows the mean power of the Pre-Stage, Pain-1, and Post-Stage for each group. The second row shows the significant difference between the Pre-Stage and CPT1 as well as Post-Stages across all groups. The third row shows the significant difference between CPT1 vs. Post-Stages for all groups. Black circles indicate reductions in significant power, and grey circles indicate increases.

Specifically, there was a significant global decrease in the alpha power in the CPT1 condition, and a significant decrease only in the occipital and parietal region was observed in the Post-Stage of G2, G3, and G4 compared to the Pre-Stage. Conversely, there was a significant decrease in the alpha power in the Post-Stages of G1 in the frontal region. When compared to the CPT1 condition, G1 and G3 significantly increased in the occipital and parietal region only, while G2 and G4 increased in the frontal, parietal, and occipital regions.

Figure 9 shows the EEG power of G1, G2, G3, and G4 within the beta-band. The first row shows the mean power of the Pre-Stage, CPT1, and Post-Stage for each group. The second row shows the significant difference between the Pre-Stage and CPT1 as well as Post-Stage across all groups. The third row shows the significant difference between CPT1 vs. Post-Stages for all groups. Black circles indicate reductions in the significant power, and grey circles indicate increases. Specifically, significant global increases in the beta power were observed for the CPT1 condition as well as following stages of G1. Beta power also increased significantly in the frontal region in both G2 and G3 in their Post-Stage as compared to Pre-Stage. But there was a significant reduction (Post-Stage) in the beta power in the frontal region observed in G4. When compared to the CPT1 condition, G2 and G3 significantly decreased in the frontal region only, while G4 globally decreased.

Figure 10 shows the EEG power for G1, G2, G3, and G4 within the gamma-band. The first row shows the mean power of the Pre-Stage, Pain-1, and Post-Stage for each group. The second row shows the significant difference between the Pre-Stage and CPT1 as well as Post-Stage across all groups. The third row shows the significant difference between CPT1 vs. Post-Stages for all groups. Black circles indicate reductions in the significant power, and grey circles indicate increases. Specifically, significant global increases in the gamma power were observed for the CPT1 condition as compared to Pre-Stage. The gamma power also increased significantly in the parietal and occipital region in the Post-Stage of G1 as compared to Pre-Stage. But there was a significant reduction (Post-Stage) in the gamma power in the same region observed in G4. When compared to the CPT1 condition, G1, G2, and G3 exhibited a significant decrease in the frontal and occipital region only, while G4 displayed a global decrease.

## 4. Discussion

The current study investigated the effectiveness of neuromodulation techniques on pain inhibition and cognitive function of attention. This study demonstrates enhancement of attention and amelioration of pain perception when tDCS and TENS are applied to healthy participants. These improvements are evident in both behavioral and neurological assessments. Behavioral improvements are observed with improved Stroop task scores for cognition enhancement, and pain inhibition is noticeable by Numeric Pain Scale ratings supported by neurological evaluation with topoplot spectral analysis.

The novel finding of this study is that anodal-tDCS to DLPFC improves attention, an important component of cognition, compared to cathodal-tDCS. In contrast, when a painful stimulus is administered, cathodal-tDCS stimulations are effective in augmenting pain inhibition. These findings are extendable to TENS, which replicates the same results by supplementing attention and elevating pain perception.

Anodal-tDCS to the DLPFC in the current study leads to an increase in Stroop task performance representing an improvement in a higher level of concentration and eventually alertness [55]. The results are congruent with previous studies highlighting improved cognitive performance and reduced confusion in different meanings of a word, implying the role of the DLPFC in cognitive function such as working memory and attention [56,57,58,59,60]. This leads to inferences in healthy individuals as well as those with clinical conditions such as Parkinson’s disease, Alzheimer’s dementia, and others where cognitive functions are compromised. The possible underlying mechanism may also be related to dopamine neurotransmission, as evidenced by studies demonstrating its enhancement in association with attentional improvement [61,62]. Studies show that anodal-tDCS may result in dopamine activation, leading to improved attention and working memory in patients [63,64]. These findings are in line with our results, indicating that anodal-tDCS stimulation of the DLPFC might increase the cognitive function level by improving the Stroop task performance and lowering the number of errors.

Significant positive influence of TENS treatment on the interference score was found during Stroop task performance. Additionally, verbal and visual short-term and long-term memory performance was sustained with peripheral stimulation in patients with cognitive impairment and memory decline [65,66]. Previous research has also confirmed the activation of the brain regions related to focus when TENS is applied to the median nerve [67,68]. Therefore, a positive impact on executive brain functions of small to moderate strength is anticipated with TENS stimulation, as it is going to mimic the activity of the prefrontal cortex and anterior cingulate cortex, which are regions responsible for attention.

An increment in the theta frequency band power spectral density alongside a decline in the alpha power spectral strength is a sign of alertness and attentiveness [69]. An increase in the theta oscillatory power is essential for cognitive information processing like learning, memory encoding, and creative activities [70,71]. This present study shows that the power spectrum is increased immediately after performing the Stroop test, which may be due to the activation of definite brain regions such as the prefrontal cortex and anterior cingulate cortex that plays an important role in cognitive functions as reported through fMRI, EEG, and PET studies [69,72,73,74]. As discussed in the present study, the same findings have been observed in other studies that have reported synchronization of theta while the alpha is desynchronized for memory demanding task [75,76].

Earlier studies argued that DLPFC is the most suitable target for exerting additional analgesic effects of tDCS [77]. In this study, the CPT (Cold Pressor Test) was applied as a tool for inducement of sensory and painful distractor or pain stimuli. To help reduce the effect of ambient temperature on the pain perception measures, temperature was held relatively stable. Participants in this study used the numerical pain scale to rate their pain level as well. A drop in subjective pain rating was recorded, reflecting the fact that after cathodal-tDCS stimulation, the subjects faced increased pain inhibition, because their perceptual threshold of pain was raised compared to them after anodal-tDCS stimulation. The results, therefore, are consistent with earlier studies carried out either on healthy individuals or on those with spinal cord injury [78,79]. In addition, these findings had corroboration from a previous study showing pain relief as a result of anodal-tDCS stimulation of the somatosensory cortex [80]. While this research showed no shift in objective pain threshold values, a past study noted no change in subjective pain threshold scores [81].

Contrary to the findings of the present study, anodal-tDCS to the DLPFC was reported to increase high thermal pain tolerance threshold. Analogously, an increment in the electrical pain threshold was observed in another study with anodal-tDCS without any comparison to cathodal-tDCS [82]. Consistent with this, an increment in time latencies to pain threshold and pain tolerance was also observed for anodal-tDCS stimulation of primary motor areas resulting in increases in the cold pain threshold and tolerance [34]. Among different regions of the brain, tDCS over motor cortex yields most favorable results for alleviation of chronic pain as indicated in a review of previous studies [83]. Anodal-tDCS of M1 also exhibited ameliorated chronic pain and so was recommended as a promising method for the treatment and management of such pain specifically as fibromyalgia, neuropathic pain and back pain [84].

In the present study, electrical stimulation with TENS is also associated with pain relief. Somatic stimulation inhibits pain signals via activation of central and peripheral mechanisms. TENS has been found to be an effective non-pharmacological treatment in patients suffering from pain [85]. An elevated ice pain threshold to relieve experimentally induced cold pain is also noted with conventional electrical stimulation by TENS [86]. TENS has been reported to be effective in relieving experimental and clinical pain by increasing pain threshold in healthy control individuals as well as in clinical pain population and can therefore be used as a complementary treatment to medication [87].

Multiple studies have been conducted that have employed CPT to study EEG rhythms in healthy individuals. In the current study, visual inspection of EEG topographs reveals a significantly raised high-frequency (beta and gamma) activity, increased low-frequency (theta) activity, and a degraded medium-frequency (alpha) activity during the CPT. These characteristics of the EEG power spectrum are in line with previous studies, which reflect an increased activity in the beta-frequency band and diminished activity in the alpha-frequency band in cold pain during a comprehensive analysis of the EEG power spectrum [88,89]. Both alpha and beta rhythms are considered to be a potential indicator of pain, where decline in the alpha power capacity and an elevated beta power intensity constitute an integral biomarker of pain. Enhanced beta activity has been found to be associated with an increased response to cold pain stimuli. Analogously, a reduction in the activity in the alpha band has been interpreted as attention processing towards a nociceptive signal [90,91]. Evidence suggests that gamma oscillation is also an indicator of pain perception and pain processing and is found to be associated with the intensity of the perceived pain. A significant enhancement of the total gamma power post-induction of pain in healthy individuals corresponds to the fact of paying more attention to pain causing stimuli [92].

A reduced gamma activity and an increased alpha and theta activity post-administration of TENS and tDCS (anodal and cathodal) is evident in the present study. This is consistent with the findings of previous studies, which indicate enhanced alpha power and reduced activity of the gamma band triggered when diverting attention from pain processing and thereby exhibiting effectiveness of neuromodulation in healthy participants [93]. The efficacy of neuromodulation methods can be explained by the brain gate theory in such a way that the electrical stimulation from these modalities delivers an even stronger input to the central nervous system compared to pain, which in turn restrains and blocks transmission and processing of nociceptive stimuli from neuron to brain.

While unveiling the dual benefits of TENS and tDCS for pain relief and cognitive enhancement in healthy individuals, this study has inherent limitations. The exclusive focus on healthy participants may limit direct applicability to clinical populations. Short-term assessments might not fully capture long-term effects, and the individual comparison of anodal-tDCS, cathodal-tDCS, and TENS leaves questions about relative effectiveness. Moreover, the duration for which the effect of neuromodulation endured was not elucidated. Furthermore, the neurological changes in the deep cortical structures and connectivity changes in brain regions related to the pain matrix and cognitive functions are not investigated [94,95,96]. Furthermore, future studies can explore the role of neuromodulation in low back pain patients on the effectiveness of pain relieving interventions such as trunk orthosis [97]. Moreover, neuromodulation can combine sensor fusion techniques with EEG, EMG, and IMU data [98] to apply machine learning for evaluating variations between an individual’s intention and performance. Notwithstanding, the findings confirm the efficacy of TENS and tDCS as possible non-drug therapeutic alternatives for cognition as well as alleviation of pain.

## 5. Conclusions

The findings confirm the efficacy of TENS and tDCS as possible non-drug therapeutic alternatives for cognition as well as alleviation of pain.

## Figures and Tables

**Figure 1 biomedicines-12-01269-f001:**
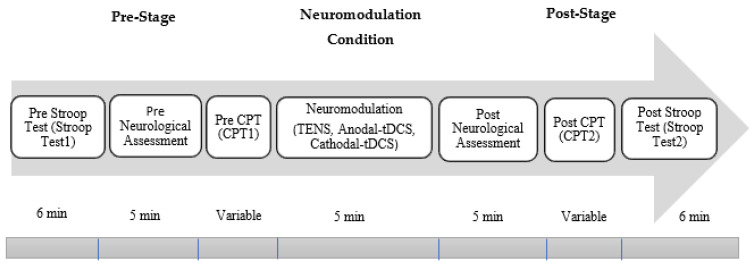
Experimental setup and protocols for EEG data collection. The experimental procedure was categorized into four groups (G1–G4), where G1 was the control and G2–G4 were subjected to special neuromodulation interventions (TENS, anodal-tDCS, and cathodal-tDCS).

**Figure 2 biomedicines-12-01269-f002:**
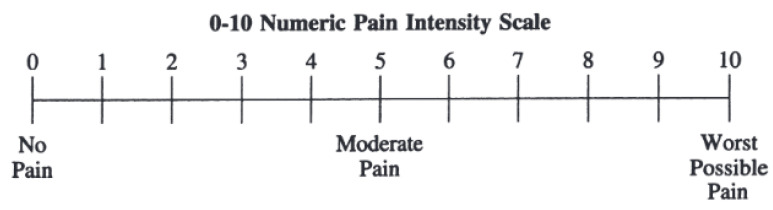
Standardized 11-point Numeric Pain Scale. A ‘0’ represents no pain, and a ‘10’ represents extremely severe pain.

**Figure 3 biomedicines-12-01269-f003:**
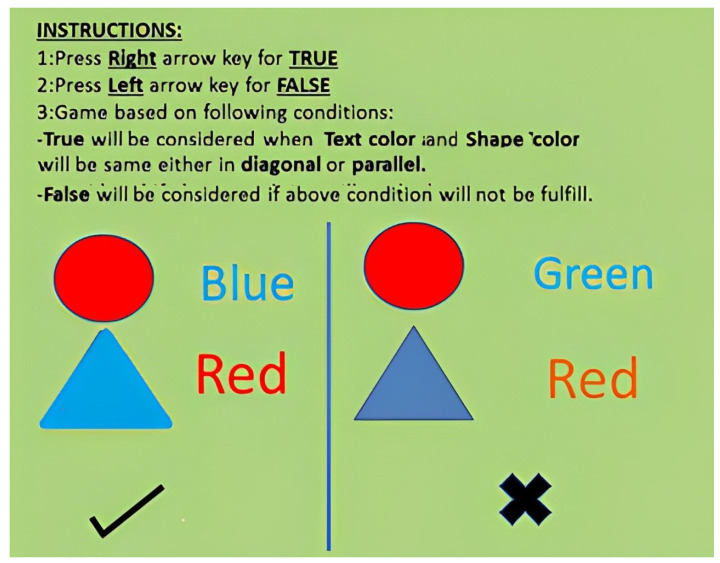
Pictorial representation of the Stroop task used in the study. For congruent pairs of geometric shape color and text color, participants pressed the right arrow; for incongruent pairs, they pressed the left one.

**Figure 4 biomedicines-12-01269-f004:**
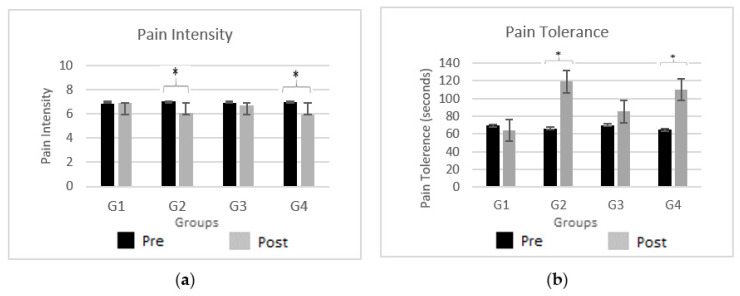
Pain Assessment. Comparison between CPT1 (black bars) and CPT1 pain assessments (gray bars) for all four groups (G1, G2, G3, and G4). Sub-figure (**a**) shows the mean pain intensity level obtained by the participants, while sub-figure (**b**) shows the mean pain tolerance level. Significant changes (*p* < 0.05) are marked by ‘*’ above the bars.

**Figure 5 biomedicines-12-01269-f005:**
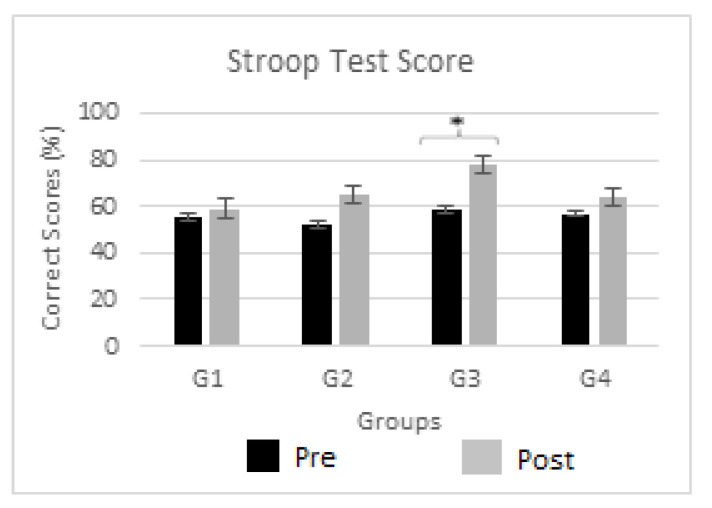
Stroop task performance. Comparison between a cognitive task (i.e., Stroop task score) for Stroop Test1 (black bars) and Stroop Test2 (gray bars) for all four groups (G1, G2, G3, and G4). Significant changes (*p* < 0.05) are marked by ‘*’ above the bars.

**Figure 6 biomedicines-12-01269-f006:**
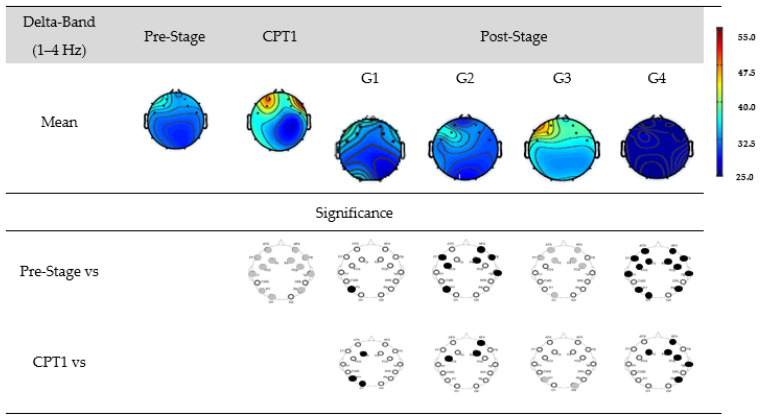
EEG power for G1, G2, G3, and G4 within the delta-band. The first row shows the mean of the Pre-Stage, CPT1, and Post-Stage for each group (G1, G2, G3, and G4). The second row shows the significant difference between Pre-Stage vs. CPT1 and Pre-Stage vs. Post-Stage across all groups. The third row shows the significant difference between CPT1 vs. Post-Stage for all groups. Reductions in significant power are denoted by black, while increases are represented by grey.

**Figure 7 biomedicines-12-01269-f007:**
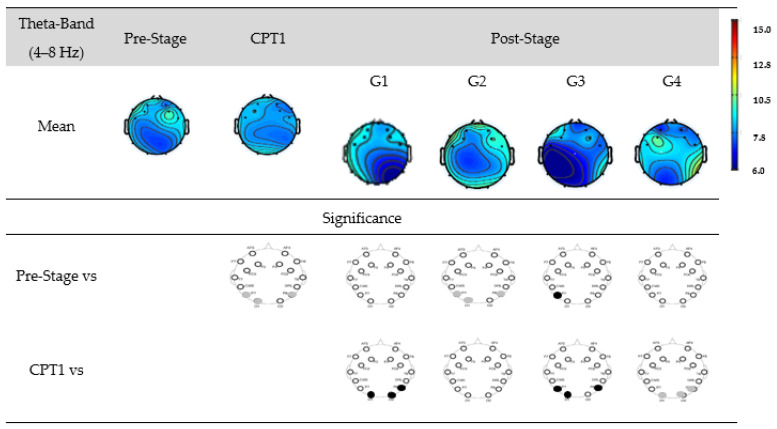
EEG power in G1, G2, G3, and G4 within the theta-band. The first row shows the mean of the Pre-Stage, CPT1, and Post-Stage for each group (G1, G2, G3, and G4). The second row shows the significant difference between Pre-Stage vs. CPT1 and Pre-Stage vs. Post-Stage across all groups. The third row shows the significant difference between CPT1 vs. Post-Stage for all groups. Reductions in significant power are denoted by black, while increases are represented by grey.

**Figure 8 biomedicines-12-01269-f008:**
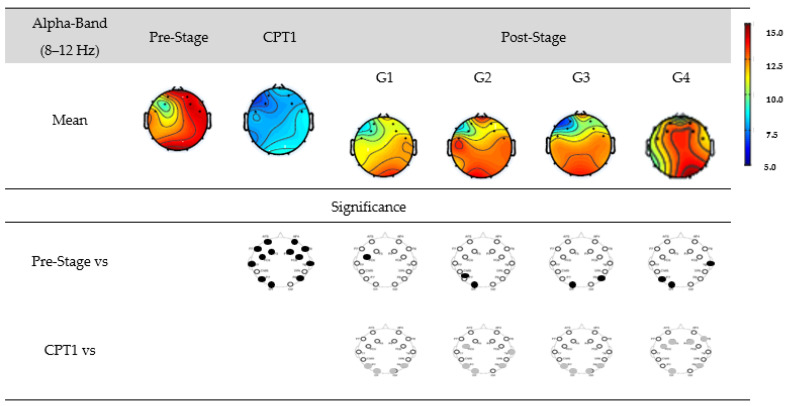
EEG power in G1, G2, G3, and G4 within the alpha-band. The first row shows the mean of the Pre-Stage, CPT1, and Post-Stage for each group (G1, G2, G3, and G4). The second row shows the significant difference between Pre-Stage vs. CPT1 and Pre-Stage vs. Post-Stage across all groups. The third row shows the significant difference between CPT1 vs. Post-Stage for all groups. Reductions in significant power are denoted by black, while increases are represented by grey.

**Figure 9 biomedicines-12-01269-f009:**
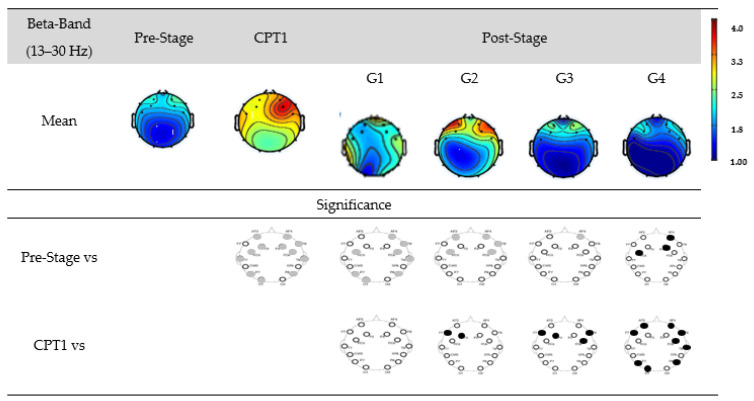
EEG power for G1, G2, G3, and G4 within the beta-band. The first row shows the mean of the Pre-Stage, CPT1, and Post-Stage for each group (G1, G2, G3, and G4). The second row shows the significant difference between Pre-Stage vs. CPT1 and Pre-Stage vs. Post-Stage across all groups. The third row shows the significant difference between CPT1 vs. Post-Stage for all groups. Reductions in significant power are denoted by black, while increases are represented by grey.

**Figure 10 biomedicines-12-01269-f010:**
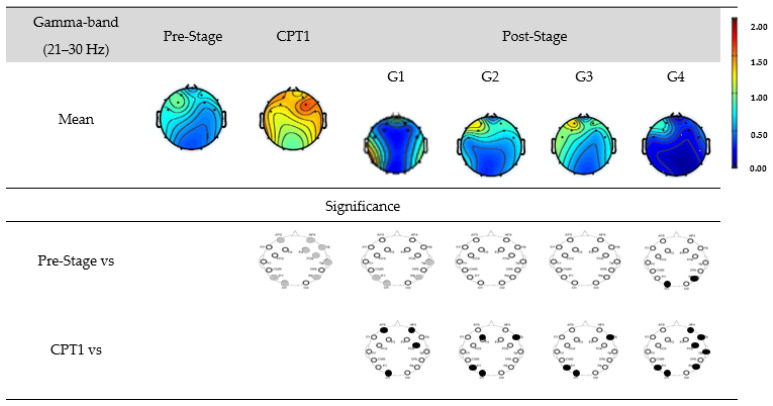
EEG power in G1, G2, G3, and G4 within the gamma-band. The first row shows the mean of the Pre-Stage, CPT1, and Post-Stage for each group (G1, G2, G3, and G4). The second row shows the significant difference between Pre-Stage vs. CPT1 and Pre-Stage vs. Post-Stage across all groups. The third row shows the significant difference between CPT1 vs. Post-Stage for all groups. Reductions in the significant power are denoted by black, while increases are represented by grey.

## Data Availability

The datasets generated and/or analyzed during the current study are not publicly available due to privacy and ethical concerns but are available from the corresponding author on reasonable request.

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
