# Peer review of "Comparative Neurological and Behavioral Assessment of Central and Peripheral Stimulation Technologies for Induced Pain and Cognitive Tasks"

_biomedicines, 2024, doi:10.3390/biomedicines12061269_

Round 1

Reviewer 1 Report

Comments and Suggestions for Authors

At the manuscript “Comparative Neurological and Behavioral Assessment of Central and Peripheral Stimulation Technologies for Induced Pain and Cognitive Tasks” by Drs. Muhammad Danish Mujib et al the authors compare effectiveness of central stimulation (tDCS) and peripheral stimulation (TENS) for pain suppression during cognitive tasks in healthy volunteers. The protocol included pain induction (CPT), tDCS/TENS administration, EEG data collection and cognitive assessments. Significant EEG changes have been demonstrated, the result showed a decrease in pain intensity with certain stimulation, and certain stimulation also improves cognitive abilities. The authors concluded that TENS and tDCS are effective as non-drug therapeutic alternatives for pain relief.

 The authors conducted a really impressive and work consuming study and obtained very interesting results. I have no objections to the essence of the study, but there are some questions.

Among the results, what interested me most was the fact that anodal-tDCS to DLPFC improves attention. It seems to me that it is worth dwelling more on this fact - that is, adding a few sentences about the biological basis of this phenomenon?

Since, as the authors write, anodal stimulation improves symptoms in Parkinson's disease, could this be due to additional dopamine release?

 Is it possible to conclude whether dopaminergic neurons entered the stimulation zone? And if so, in what structures of the brain?

 Minor criticisms:

LINE 159 2.5. Transcranial Electrical Nerve Stimulation

LINE 163 electrodes of 5 cm x 5 cm. Both electrodes were positioned over the median nerve….

There is probably a typo here: if the median nerve on the hand was stimulated, the stimulation cannot be transcranial.  

It also makes sense to introduce justification for the stimulation parameters: knowing the current strength and the approximate distance of the nerve from the electrode, it can be approximately proven that the nerve was stimulated enough and not too much. I would suggest using reference, for example: DOI: 10.1364/ol.33.001032

The presentation of a subject is systematic and comprehensive and analysis is proper. I am happy to recommend the manuscript for the publication after minor corrections mentioned above.

Author Response

Reviewer 1:

S.No

Reviewers comments

Reply

1

Among the results, what interested me most was the fact that anodal-tDCS to DLPFC improves attention. It seems to me that it is worth dwelling more on this fact - that is, adding a few sentences about the biological basis of this phenomenon?

Since, as the authors write, anodal stimulation improves symptoms in Parkinson's disease, could this be due to additional dopamine release?

 Is it possible to conclude whether dopaminergic neurons entered the stimulation zone? And if so, in what structures of the brain?

Information are added. Refer line 435-441

2

LINE 159 2.5. Transcranial Electrical Nerve Stimulation

LINE 163 electrodes of 5 cm x 5 cm. Both electrodes were positioned over the median nerve….

 There is probably a typo here: if the median nerve on the hand was stimulated, the stimulation cannot be transcranial.  

Yes it was a typo error. Now it is corrected (Refer line 167)

3

It also makes sense to introduce justification for the stimulation parameters: knowing the current strength and the approximate distance of the nerve from the electrode, it can be approximately proven that the nerve was stimulated enough and not too much. I would suggest using reference, for example: DOI: 10.1364/ol.33.001032

References are added (Refer line 174)

Reviewer 2 Report

Comments and Suggestions for Authors

The article entitled "Comparative neurological and behavioural assessment of central and peripheral stimulation technologies for induced pain and cognitive tasks" studies the effect of TENS and tDCS on healthy volunteers. The idea is to induce pain in the dominant hand immersed in increasingly cold water (Cold Pressor Test) and to assess the level of pain and attention in different groups of patients: G1: control patients, G2: patients treated with TENS placed at the wrist of the hand immersed in cold water, G3: patients treated with anodal tDCS (normally on the opposite side to the immersed hand) targeting the dorsolateral prefrontal cortex, G4: same as in G3 but with cathodal stimulation.

The subject is interesting, but I see many imprecisions and errors in the choice of protocol, which makes the results very difficult to interpret, if not uninterpretable.

Abstract:

tDCS and TENS must be defined.

The target of tDCS must be mentioned.

CPT must be defined.

Everyone knows the p-value, but t-value needs to be defined.

EEG-data should be summarized in a synthetic way showing their interest.

Introduction: 

What does AIN mean? 

Line 56: your interpretation of the Gong W 2019 article (ref 11) is wrong. Pain does not degrade all aspects of attention. The executive aspects of attention are not altered.

Line 79: Your interpretation of Boggio PS 2008 (ref 31) is wrong. Pain circuits do not only involve the dorsolateral prefrontal cortex but primarily the motor cortex (M1) which should have been chosen as the target for tDCS.

Line 97: The side stimulated by tDCS (right or left) should be mentioned.

Line 98: A reference is needed for the Cold Pressor test.

Line 100: A reference is needed for the stroop test, in particular the reference validating the test in Pakistani. The version of the stoop test used should also be mentioned.

Marerials and methods:

Line 103: "80" is unnecessary.

Line 111: The reference of the document mentioning the agreement of the ethics committee and the name of the university which issued this document must be given.

Line 122: Four groups.

Line 127: It is not clear why only 12 of the 20 patients in each group underwent neurological assessment. 

Line 131: This is the main criticism. The dorsolateral prefrontal cortex (DLPFC) corresponds to F3 (left) or F4 (right). FP1 and FP2 correspond to the supraorbital cortex, which is supposed to be functionally "neutral". If these are the targets targeted in tDCS, the work has virtually no value. Moreover, tDCS sessions usually last 20 minutes. A 5-minute tDCS session can practically be considered as a "sham". The sitimulated sides are not specified. One might think that the target is on the right, since it is the left hand that is immersed in the cold water.

Line 164: It can be assumed that the TENS were placed on the wrist above the hand immersed in cold water. This should be clarified.

Given the doubt about the targets stimulated by tDCS, I do not see what this study, which required a great deal of work, can contribute scientifically. The work could be reconsidered if we were certain that it was indeed the CPFDL that was stimulated and that the side was specified. Personally, I don't see the point of cathodal stimulation. It would have been better to have a group of "sham" tDCS and in any case to stimulate for 20 minutes.

Author Response

Reviewer 2:

S.No

Reviewers comments

Reply

1

Abstract:

tDCS and TENS must be defined.

The target of tDCS must be mentioned.

CPT must be defined.

Everyone knows the p-value, but t-value needs to be defined.

EEG-data should be summarized in a synthetic way showing their interest.

Defined (Refer line 22-23)

Defined (Refer line 22-23)

Defined (Refer line 26)

t-value defined (Refer line 30-31)

Condensed the EEG data summary in the abstract to effectively highlight key findings. (Refer line 34-35)

2

Introduction:

What does AIN mean?

It was a typo error. Actually it was PAIN. It is corrected now. (Refer line 47)

3

Line 56: your interpretation of the Gong W 2019 article (ref 11) is wrong. Pain does not degrade all aspects of attention. The executive aspects of attention are not altered.

Replaced ref 11 (Gong, study) with a more relevant study (Moore study) (Refer line 61)

4

Line 79: Your interpretation of Boggio PS 2008 (ref 31) is wrong. Pain circuits do not only involve the dorsolateral prefrontal cortex but primarily the motor cortex (M1) which should have been chosen as the target for tDCS.

Replaced ref 31 (Boggio, study) with a more relevant study (Zoha Deldar study). (Refer line 85)

5

Line 97: The side stimulated by tDCS (right or left) should be mentioned.

Information are added. (Refer line 100-101)

6

Line 98: A reference is needed for the Cold Pressor test.

References are added. (Refer line 103)

7

Line 100: A reference is needed for the stroop test, in particular the reference validating the test in Pakistani. The version of the stoop test used should also be mentioned.

References are added. (Refer line 104)

8

Materials and methods:

Line 103: "80" is unnecessary.

We conducted an interim analysis to determine effect size, statistical power, and sample size, leading us to select 80 participants for the study. (Refer Result section line 255 - 265).

9

Line 111: The reference of the document mentioning the agreement of the ethics committee and the name of the university which issued this document must be given.

The study was conducted with the approved of Research Ethic Committee, NED University of Engineering & Technology. Refer line 113-114

10

Line 122: Four groups.

The information is correct. Refer line 127-128

11

Line 127: It is not clear why only 12 of the 20 patients in each group underwent neurological assessment.

Mentioned in result section from line 270- 275. (All 80 participants involved …..with no persistent negative effects noted)

12

Line 131: This is the main criticism. The dorsolateral prefrontal cortex (DLPFC) corresponds to F3 (left) or F4 (right). FP1 and FP2 correspond to the supraorbital cortex, which is supposed to be functionally "neutral". If these are the targets targeted in tDCS, the work has virtually no value. Moreover, tDCS sessions usually last 20 minutes. A 5-minute tDCS session can practically be considered as a "sham". The sitimulated sides are not specified. One might think that the target is on the right, since it is the left hand that is immersed in the cold water.

The Emotiv EPOC 14-channel EEG lacks Fp1 and Fp2 electrodes. It was a typographical error. We actually used F3 and F4 for anodal and cathodal tDCS stimulation, respectively. Now we corrected. Refer line (159-164).
Moreover, 10 minutes of sham stimulation, followed by an additional 5 minutes of active stimulation were provided, allowing participants to acclimate to the sensation. Refer line 122-123.

13

Line 164: It can be assumed that the TENS were placed on the wrist above the hand immersed in cold water. This should be clarified.

Clarification added in the mentioned section. (see line 171-174)

Reviewer 3 Report

Comments and Suggestions for Authors

Dear Authors,

I read with interest your paper since it tries to deepen the effects of central and peripheral stimulations on cognitive tasks and pain.

Nevertheless, there are many concerns and flaws which make it not suitable for publication in this Journal.

First of all, English should be globally revised.

Both in the abstract and in the method section you did not describe the study model. Moreover, the study involvedshumans and it lacks a specific approval from the local ethical committee. You made a group comparison, but you did not provided a sample size calculaton. On which basis did you establish that eighty participants were a correct number?

As a consequence, the results are not reliable and they make your conclusions weak and not solid.

Moreover, the paper layout changes in the passage between results and discussion sections.

I appreciate your efforts but in my opinion this paper should be globally reconsidered before a further submission.

Best regards

Comments on the Quality of English Language

The quality of english is scarce.

Just as an example, at lines 21-23, it is written: "This study compares the effectiveness of central-stimulation (tDCS) and peripheral-stimulation (TENS) in pain-inhibition during a cognitive-task in healthy volunteers and to observe potential neuro-cognitive improvements. Healthy eighty volunteers went through". Why did you insert "and" between volunteers and to? Heathy should follow the word eighty.

So, I suggest a moderate editing.

Author Response

Reviewer 3:

S.No

Reviewers comments

Reply

1

English should be globally revised.

We have thoroughly reviewed the manuscript, and we have revised and improved the English in the Introduction (refer line 61-64, 72-81, 93-96), Materials and Methods (refer line 108-111,116-128-276), results (refer line 255-275), discussion sections (428-472) and conclusion (refer line 533- 534).

2

Both in the abstract and in the method section you did not describe the study model.

A brief overview of the study model is provided in the abstract (refer line 25-28), while a comprehensive explanation is given in the method section Refer line (refer line 116-126)

3

Moreover, the study involved humans and it lacks a specific approval from the local ethical committee.

The study was conducted with the approved of Research Ethic Committee, NED University of Engineering & Technology. (Refer line 113-114)

4

You made a group comparison, but you did not provided a sample size calculation. On which basis did you establish that eighty participants were a correct number?

As a consequence, the results are not reliable and they make your conclusions weak and not solid.

We conducted an interim analysis to determine effect size, statistical power, and sample size, leading us to select 80 participants for the study. (Refer Result section line 255 - 265).

5

Moreover, the paper layout changes in the passage between results and discussion sections.

Corrected

Reviewer 4 Report

Comments and Suggestions for Authors

The present study aims to compare the effectiveness of central stimulation (tDCS) and peripheral stimulation (TENS) in pain inhibition during a cognitive task in healthy volunteers and to observe potential neurocognitive improvements. The results highlighted the efficacy of TENS and tDCS as possible non-drug therapeutic alternatives for cognition with alleviation from pains.  

This is a well-written paper. The manuscript is well structured, and the primary purpose is clear. Methods are adequate, and the data are clearly shown. The data support conclusions. All the factors mentioned above are described clearly. However, I have several suggestions. 

- Line 43: AIN - should be Pain?

- Line 49 - [4]–[7]. please adjust the citation style to the journal's requirements

-  Line 102: The inclusion and exclusion criteria could be more precise. Did patients feel any pain at the time of recruitment? Were other general diseases excluded?

- Line 124: Please provide the gender and age distribution in individual study groups.

-       Please improve the style, language, and interpunctions in the paper.

In general, the work is interesting and can contribute to the literature. I hope my suggestions will help improve this work.

Comments on the Quality of English Language

Minor editing of English language required

Author Response

Reviewer 4:

S.No

Reviewers comments

Reply

1

- Line 43: AIN - should be Pain?

It was a typo error. Actually it was PAIN. It is corrected now. (Refer line 47)

2

Line 49 - [4]–[7]. please adjust the citation style to the journal's requirements

Corrected as per MDPI recommended citation style (ACS style)

3

-  Line 102: The inclusion and exclusion criteria could be more precise. Did patients feel any pain at the time of recruitment? Were other general diseases excluded?

The inclusion and exclusion criteria have been specified precisely. Refer line 108-111

4

Line 124: Please provide the gender and age distribution in individual study groups.

Refer line 266-269

Round 2

Reviewer 3 Report

Comments and Suggestions for Authors

I really appreciated the efforts to improve the quality of your paper according to my suggestions. Many of the criticities have been solved, just a methodological aspect has to be further considered. If it is a trial deepening a comparison between groups, is it a randomized trial? If yes, please clarify this aspect in the method section and provide a clinical trial registration.

Best regards

Author Response

Reviewer 3 Comments (2nd Round):

S.No

Reviewers comments

Reply

1

If it is a trial deepening a comparison between groups, is it a randomized trial? If yes, please clarify this aspect in the method section and provide a clinical trial registration.

We want to clarify that our study is conducted on healthy subjects rather on patients. Therefore, it is not a clinical trial or a randomized controlled study.